# A Movement Classification of Polymyalgia Rheumatica Patients Using Myoelectric Sensors

**DOI:** 10.3390/s24051500

**Published:** 2024-02-26

**Authors:** Anthony Bawa, Konstantinos Banitsas, Maysam Abbod

**Affiliations:** Department of Electronic and Electrical Engineering, Brunel University London, Uxbridge UB8 3PH, UK

**Keywords:** gait disorder, polymyalgia rheumatica, classifiers, pattern

## Abstract

Gait disorder is common among people with neurological disease and musculoskeletal disorders. The detection of gait disorders plays an integral role in designing appropriate rehabilitation protocols. This study presents a clinical gait analysis of patients with polymyalgia rheumatica to determine impaired gait patterns using machine learning models. A clinical gait assessment was conducted at KATH hospital between August and September 2022, and the 25 recruited participants comprised 18 patients and 7 control subjects. The demographics of the participants follow: age 56 years ± 7, height 175 cm ± 8, and weight 82 kg ± 10. Electromyography data were collected from four strained hip muscles of patients, which were the rectus femoris, vastus lateralis, biceps femoris, and semitendinosus. Four classification models were used—namely, support vector machine (SVM), rotation forest (RF), k-nearest neighbors (KNN), and decision tree (DT)—to distinguish the gait patterns for the two groups. SVM recorded the highest accuracy of 85% among the classifiers, while KNN had 75%, RF had 80%, and DT had the lowest accuracy of 70%. Furthermore, the SVM classifier had the highest sensitivity of 92%, while RF had 86%, DT had 90%, and KNN had the lowest sensitivity of 84%. The classifiers achieved significant results in discriminating between the impaired gait pattern of patients with polymyalgia rheumatica and control subjects. This information could be useful for clinicians designing therapeutic exercises and may be used for developing a decision support system for diagnostic purposes.

## 1. Introduction

Gait disorder originates from motor weakness, neurological and musculoskeletal changes which are quite common among the elderly or due to sports injury [1]. Muscular diseases may create mobility limitations of the lower limb joints in the hip, knee and ankle. Polymyalgia rheumatic (PMR), a common musculoskeletal disease, is prevalent among elderly people usually from the age of 50 years and above [2]. This disease is normally characterized by stiffness of muscles around the shoulders and pelvic girdle. Furthermore, the disease is associated with pain and weakness of muscles that may impair their function. It is unknown whether the strained hip muscles of polymyalgia rheumatica patients create mobility limitations, hence the importance of conducting gait analysis. Human gait analysis is used in medical settings for prosthetic purposes [3,4]. Gait analysis plays an integral role in the treatment plan by providing useful information for rehabilitation protocol design. In addition, gait analysis allows for the investigation of sensory–motor interaction to understand the locomotor system functioning [5]. With the emergence of machine learning techniques, automated gait classification [6,7] became an evolving research area in developing smart healthcare systems. Gait analysis can be conducted either with motion capture systems or wearable sensors. Motion capture (mocap) systems were previously used to capture high-quality kinematic data of joint motion trajectory for gait classification, but they were faced with some challenges due to the complexity of the equipment settings and the cumbersome process of preparation. Furthermore, mocap systems did not provide adequate spatiotemporal parameters and convenience for monitoring daily gait activities. In recent times, the use of wearable sensors has emerged as an alternative to 3D motion capture systems for gait analysis, which is less expensive and suitable for clinical settings.

Some studies have demonstrated the use of wearable sensors as an effective tool for gait analysis [8]. Nardo et al. [9] presented the use of occurrence frequency from an electromyography (EMG) sensor to detect gait disorder. The results indicated that EMG signals were suitable for evaluating muscle activity in detecting gait abnormality. Yao Guo et al. [10] presented the use of EMG sensors to detect gait abnormality with multiclass classifiers. The results showed that SVM and BiLSTM had accuracies of 92.20% and 87.85%, respectively. Mohd et al. [11] illustrated gait abnormality for children with autism by classifying the walking pattern of autistic and normally developed children. Chakraborty et al. [12] demonstrated the detection of gait disorder using inertial sensors. Deep neural networks were used to identify abnormal gait patterns in individuals. The results indicated an accuracy of 90.97% and 96.40% subject-wise and segment-wise, respectively.

Muscle synergy in the context of gait analysis refers to the coordinated activation patterns of muscles working together to perform a specific movement, such as walking. Muscle synergy analysis provides a framework for understanding the coordination of muscle activation patterns during gait exercise, but it does not directly quantify the force produced by individual muscles. Some studies have used this approach in the analysis of gait impairement. A study by Fei et al. [13] investigates lower-limb muscle synergies in children with cerebral palsy (CP) compared to healthy adults. Surface electromyography (sEMG) signals from eight muscles were collected during walking. Muscle synergies were extracted using nonnegative matrix factorization. The results showed that there were variations detected in muscle synergy between legs in CP children, suggesting a role in walking impairment. These differences may contribute to motor impairment and reduced walking performance in CP children. Another study by Wang et al. [14] outlines a gait analysis framework for children with CP using EMG and acceleration (ACC) signals. The ACC signals were processed for stride cycle detection and kinematic information, while EMG signals assessed abnormal muscle activation patterns. The results demonstrate the method’s potential for assessing and treating lower extremity functions in CP children. This was useful in detecting gait abnormality based on the muscle activation pattern for children with CP. The work by Barroso et al. [15] integrates instrumented gait analysis and muscle synergy analysis to improve the assessment of post-stroke gait deficits. The main findings of this work identified predictive variables, including kinematic, kinetic, and synergy-related features, surpassing traditional assessment methods. These findings suggest personalized rehabilitation strategies targeting biomechanical and neural control features for enhanced walking function in post-stroke patients.

Currently, from the reviewed literature and to the best of our knowledge, there is a deficit of information regarding how intrinsic muscle activity affects the movement of patients with polymyalgia rheumatica. The fundamental difference in investigating the strained hip muscles in the movement of PMR patients compared with other musculoskeletal diseases in previous works lies in the unique pattern of pain and stiffness associated with the hip muscles of PMR patients. It is uncertain if the strained hip muscles affect the mobility of patients. Therefore, the main aim of this study is to present a clinical gait analysis of polymyalgia rheumatica patients to identify gait impairments based on hip muscle activation using machine learning models. The muscle synergy of patients would provide a clear understanding of the movement pattern. The classification of the movement gives valuable insights into the hip muscle activity and function of patients in a gait cycle.

## 2. Materials and Methods

### 2.1. Participants

In total, 25 participants, including 18 patients with polymyalgia rheumatica and 7 control subjects, were recruited from Komfo Anokye Teaching Hospital (KATH) in Ghana, between August and September 2022. The patients included 12 females and 6 males, while the control subjects included 5 females and 2 males. The demographic representation of the participants follows: age 56 years ± 7, height 175 cm ± 8, and weight 82 kg ± 10. Ethical approval for the study was given by Brunel University London, which complies with the Helsinki Declaration. Participants were given a consent form to sign before taking part in the study.

### 2.2. Testbed for Data Collection

A Delsys Trigno Avanti electromyography sensor was used for muscle signal data collection. The hip muscles measured were rectus femoris (RF), vastus lateralis (VL), biceps femoris (BF), and semitendinosus (SE). These were the strained hip muscles affected by polymyalgia rheumatica, which is the focus of the study. The hair on the hip was shaved and cleaned to give the sensors better conductivity and more accurate readings. Sensors were placed on the right and left hips of the participants concurrently to record muscle signals. Each participant was asked to perform four gait trials at normal walking speeds in a straight walkway of 10 m. Figure 1 illustrates the human gait cycle which normally alternates between the stance phase to the swing phase of a person in motion [16]. The figure depicts the EMG data recorded from the hip muscles and a participant conducting the gait exercise. In total, 100 gait cycles using four sensor channels were collected from patients and healthy controls for the classification. This was sufficient for the classification problem. Using a small number of patients EMG datasets for classification can be justified by the ability to select a reduced set of discriminating features. Studies in [17,18] have shown that EMG classification problems can achieve high accuracy even with a small number of patients. Furthermore, the automatic classification of normal and abnormal EMG patterns using machine learning algorithms is possible with high accuracy even when targeting a small number of patients. Nazmi et al. [19] used 23 participants in the classification of stance and swing phase with EMG data. Therefore, working with a small number of patients can reduce EMG data variability, which is beneficial for classification tasks while still maintaining high classification accuracy.

### 2.3. EMG Data Recorded and Processing

The raw EMG data recorded were represented as $Y_{z}\left(r\right)$, where r ∈[1,R] and R represents the total frames of the EMG data sequences, while z ∈[1,Z], and Z indicates the number of sensor channels in a repetitive gait cycle. In the gait cycle, the sequence $Y_{z}\left(r\right)$ is first segmented into a single K gait cycle $\left\{w^{1}\left(r\right)\ldots w^{k}\left(r\right)\ldots w^{K}\left(r\right)\right\}$. The raw EMG signal represents the electrical activity of muscles during contraction and relaxation shown in Figure 2. The signal is complex and time-varying which reflects the recruitment and firing of motor units within the hip muscles. The raw EMG signal recorded contains some noise that requires filtring. A Butterworth filter was used between 20 and 450 Hz, which best displays the muscle function of the muscle and removes powerline noise [20].

After filtering, the EMG signal was rectified to emphasize the absolute amplitude of the signal by removing negative values. This was achieved by taking the absolute value of each data point in the EMG signal. Rectification is a common pre-processing procedure used in EMG signal analysis. The rectification of EMG signals is important for enhancing the timing information and firing rate characteristics of the underlying motor unit action potentials. The RMS of a signal is a measure of the average power present in the signal. RMS is a fundamental metric in signal processing that can be used for pattern recognition and classification. Figure 3 shows the rectified EMG signal integrated with the RMS of the signal.

The Motor Unit Action Potentials (MUAPs) from EMG signals are generated by a group of muscle fibers that are innervated by a single motor neuron. The motor unit parameters of EMG signals vary depending on the muscles being measured. Patients with musculoskeletal disorders may suffer muscle denervation from the motor unit to the muscle fiber. The denervation of muscles can lead to the loss of nerve supply to these muscles, which results in a decrease in MUAP amplitude with an increased duration [21]. To determine muscle denervation with motor unit action potential, the key parameters used are amplitude and duration. The MUAP amplitude and duration were used to determine the size of the motor unit [22], which is essential to distinguish the EMG signals between healthy control subjects and PMR disease.

### 2.4. Feature Extraction from EMG Data

The muscle signals were extracted in the time domain (TD) and frequency domain (FD). The metadata of the extracted features are represented in Table 1 [23].

### 2.5. Principal Component Analysis

Principal component analysis (PCA) was applied to the processed EMG data for the classification. PCA is a non-supervised feature extraction and selection technique used in performing dimensionality reduction tasks to avoid overfitting [24]. It is used as an orthogonal transformation in explanatory data analysis and for examining the relationship between variables. By using PCA, it is possible to find a linear projection of the extracted EMG features in a lower dimensional space by reducing the cumulative error and maximizing the cumulative variance.

### 2.6. Classifier Description

Four classification models—namely, support vector machine (SVM), decision tree (DT), rotation forest (RF), and k-nearest neighbor (KNN)—were used in this study. A flow chart of the schematic diagram for the gait classification process is illustrated in Figure 4. The diagram illustrates the sequence of steps starting from collecting EMG data from participants, followed by signal pre-processing to eliminate noise, extracting features, and reducing dimensionality using PCA. Subsequently, machine learning models are trained using this processed dataset to assess their classification performance.

Support Vector Machine: SVM is the non-probabilistic binary linear classifier with an optimal hyperplane that splits data points from one class and those of other classes [25]. SVM can be used as a multi-pattern recognition of gait segments of the recorded dataset. The hyperparameters in the SVM classifier are the kernel function, regularization parameters, and class weights. In training the EMG dataset, different kernels were used to decide on the best suited one. In the SVM model, the kernel function is the most important parameter to transform the input data into higher dimensional space, where it becomes easier to separate different classes using a hyperplane. The linear kernel provides the best performance among the kernels. It is the fastest and most accurate kernel in the classification. A cross-validation was employed to estimate the classification accuracy and identify the optimal value of the regularization parameter C. The class weights were used to minimize misclassifications of each class to support the imbalanced classes in the dataset. In SVM, given a set of instances as $m_{i}$ and its categories as $n_{i}$ and $n_{i}\pm 1$, the splitting plane is given by 〈p,m〉+c=0, where $m_{i}R_{d}$ and $p_{i}R_{d}$ are the normal vectors of the hyperplane. The optimal hyperplane $n_{i}\left(\langle p,m_{i}\rangle +C\right)>1$ is decided by minimizing 〈p,p〉/2. In the case of linear non-separable data, SVM uses soft edges to minimize the additional slack variables $w_{i}$ and penalty parameter D. This is given by $\frac{1}{2}\langle p,p\rangle +D\sum_{i}\omega_{i},s.t\left(\langle p,m_{i}\rangle +c\right)\geq 1-\omega_{i}$ and $\omega_{i}\geq 0$.

K-Nearest Neighbors: KNN is a non-parametric classification method in which each member of a validation case is given a class label based on the voting power of the ‘K’ nearest neighbors determined by a distance measure. KNN classifiers are based on the distance function, such as the Euclidean distance for a given pair of data points in a dataset. The key parameters used in KNN for the classification include the number of neighbors and the distance metrics. The number of nearest neighbors are consided when making predictions for a new data point. This parameter determines the size of the neighborhood around each data point and has a significant impact on the model’s performance. A smaller value of K would lead to more flexible decision boundaries. With this classifier, the value of K was set to 2, indicating that two nearest neighbors were utilized in the classification process. The fine tuning of this parameter helps to optimize the performance of KNN for the classification tasks. Therefore, the number of nearest neighbors is an essential parameter in the KNN model.

Decision Tree: A decision tree (DT) classifier is an effective algorithm used in extracted feature classification in the form of a tree structure. It works by recursively partitioning the dataset into smaller subdivisions of branches and nodes. The key parameters used in decision trees for classification include the maximum depth and sample split. DT can consider a subset of features when making a split at each node. This parameter controls the maximum number of features to consider for the best split in the classification. The maximum depth parameter controls the maximum depth of the decision tree and captures the complex relationships in the EMG dataset. A maximum depth helps to prevent overfitting and performance. The minimum samples split parameter specifies the minimum number of samples required to split an internal node. In the DT classifier, the dataset was split into four levels of depth for classification. The depth of a DT is an important hyperparameter that requires precise tuning to ensure optimal performance. By fine-tuning this parameter, it helps to optimize the model for EMG classification to determine the correlation between activity and a specific condition.

Rotation Forest: The rotation forest (RF) algorithm is utilized to create classifier ensembles through feature extraction, resembling the random forest but offering advantages in overcoming certain challenges inherent in the random forest algorithm [26]. This method employs ensemble learning principles, combining feature extraction via rotation with decision trees. Key parameters in RF classification include the number of base classifiers and the size of the feature subset. With RF, the number of base classifiers, represented by the number of trees used in classification, was set to 10. Employing multiple base classifiers allows for optimal model performance, potentially resulting in enhanced classification accuracy. This parameter is important in the classification process using this model. The composition of the EMG dataset subset can vary based on the extracted EMG features used for classification. In this study, a smaller feature subset size was selected to enhance computational efficiency and mitigate overfitting. Ensemble aggregation techniques, such as majority voting and weighted averaging, are employed to amalgamate predictions from the base classifiers. This algorithm is chosen for classification due to its efficacy in improving classifier performance when dealing with limited EMG datasets compared to random forest. Additionally, RF fosters feature diversity by rotating the feature space and generating multiple feature subsets. This characteristic proves advantageous in classification tasks, where capturing diverse muscle activation patterns is important for accurately distinguishing between different classes.

### 2.7. Training and Validations of Classifiers

The EMG feature used to model the dataset for classification was the root mean square (RMS). The RMS represents the magnitude of the EMG signal and provides information about muscle activation levels. It is a commonly used feature for quantifying muscle activity in EMG classification. In this study, subject-wise splitting was used to split the dataset for training and testing. This approach ensures that the model is trained and evaluated on EMG data captured from participants. This provides a more realistic assessment of a generalized performance. Subject-wise splitting is commonly used when the goal is to evaluate how well a classification model performs across different individuals. This ensures that the model is tested on completely unseen subjects, which is particularly useful for evaluating generalization to new individuals. Each of the classifiers used in this work leveraged specific features to discern patterns in the EMG dataset, ultimately aiding in the identification of correlations between muscle activity and specific conditions. A cross-validation with stratified sampling was used for the classification in this study. For each of the classifiers, 80% of the gait cycle segments were selected for training while 20% was used for testing. A 10-fold cross-validation was applied to ensure that the overall performance variance was not overfitting for some test cases. The iteration was repeated 10 times in the training and testing for each model used. This approach provided a more robust estimate of the machine learning model performance based on the dataset. It also helped with reducing the variance of the performance for each model. To evaluate the performance of each classifier, the accuracy, sensitivity, and specificity were computed.
(1)$$Accuracy=\frac{TP+TN}{TP+TN+FP+FN}$$
(2)$$Sensitivity=\frac{TP}{TP+FN}$$
(3)$$Specificity=\frac{TN}{TN+FP}$$
where TP=truepositive, FP=falsepositive, TN=truenegative and FN=falsenegative.

## 3. Results

In this study, four classifiers were used on the EMG dataset to distinguish between normal and impaired gait patterns. The performance for each of the machine learning algorithms was computed using scikit-learn python. SVM had the highest accuracy of 85% with a sensitivity of 92%. RF had an accuracy of 80% and a sensitivity of 86%, while KNN had an accuracy of 75% with a sensitivity of 84%. DT had the lowest accuracy of 70% with a sensitivity of 90%. In terms of specificity, SVM and KNN had 71% and 57%, respectively. RF had a specificity of 67%, while DT recorded the lowest specificity of 50%. The performance for each of the classifiers is illustrated in Table 2. Further results obtained from the confusion matrix for each classifier are as follows:SVM: The false positive rate was recorded to be 28% and the false negative rate was 7%.KNN: With the KNN algorithm, the false positive rate was recorded to be 42% and the false negative rate was 15%.DT: With this classifier, the false positive rate was recorded to be 50% and the false negative rate was 10%.RF: The false positive rate for this classifier was 33% and the false negative was recorded to be 14%.

The confusion matrix serves as a crucial tool in assessing the effectiveness of classification models. It offers a comprehensive summary of the machine learning model’s performance in predicting category labels for individual input instances in the classification. Essentially, it outlines the correspondence between predicted and actual class labels assigned by the classifiers. In Figure 5, the confusion matrix was structured to index control subjects as 0 and patients as 1. Through visualizing the confusion matrix plots, it provides insight to accurately predicted cases and the instances where misclassifications occurred regarding gait impairments. This visual representation allows us to scrutinize the errors made by each classifier in distinguishing between the gait patterns of the two groups. Analyzing these discrepancies is useful for improving the classification models to enhance their accuracy and effectiveness in distinguishing between the gait of control subjects and patients. In Figure 5, the confusion matrix plots for each of the machine learning models used in the classification are displayed in predicting the gait patterns. As mentioned previously, 100 gait cycles were collected with 80 gait cycles for training and 20 for testing.

Table 3 highlights significant shifts in maximum muscle activity within the EMG signals of patients, which are attributable to factors like muscle fatigue and diminished functional capacity. Understanding muscle function during the gait cycle relies heavily on discerning the onset and offset of EMG signals. Typically, muscle activity in the gait cycle exhibits distinct phases with an onset (maximum) and offset (minimum) phase. Determining the phase of maximum muscle activity involves identifying the point at which muscle activation peaks (onset) and begins to decline (offset) relative to the gait cycle. The EMG plot in Table 3 (a) indicates that patients experience delays in onset and early offset timing of muscle activation compared to healthy individuals, who do not display such temporal discrepancies in EMG signal. These deviations may affect muscle activation timing, leading to differences in onset and offset timing between patients and healthy controls. This disparity in EMG signal timing unveils altered hip muscle activation patterns in patients, possibly stemming from hip muscle weakness, joint stiffness, and disease-associated pain. Delayed onset with early offset of muscle activation may serve as a compensatory mechanism to counter muscle weakness or fatigue during movement. Additionally, Table 3 illustrates irregular onset and offset patterns in patients compared to healthy controls, which was likely due to the disrupted normal sequence and coordination of muscle activation in the gait cycle. Irregular muscle activation patterns such as prolonged activation relative to the gait cycle may contribute to this irregularity.

Moreover, Table 3 delineates the EMG signal patterns during the gait cycle for both patients and healthy control subjects, distinguishing between stance and swing phases. The plot for each of the hip muscles illustrates the phase in the gait cycle. Patients exhibit reduced EMG signal amplitudes compared to healthy controls, suggesting possible muscle weakness. The extent of amplitude reduction in EMG signals may vary across affected muscles or muscle groups. Specifically, the four hip muscles show lower amplitude in patients compared to healthy controls during the gait cycle.

Automating the gait patterns from the EMG dataset was crucial for classifying normal and impaired movements given the challenges in visually discerning between the two groups. Figure 6 illustrates the impaired gait pattern of patients, while Figure 7 depicts the normal gait pattern of control subjects. These distinct gait patterns facilitated differentiation between the movements of the two groups.

Utilizing principal component analysis of the EMG dataset enables the automated identification of shared temporal patterns among two groups. Figure 7 depicts the gait pattern of healthy control subjects exhibiting high amplitude values that gradually decrease over time, with the x-axis denoting time (seconds) in the gait cycle and the y-axis representing amplitude (millivolts). In healthy control subjects, variations in hip muscle activation patterns over time are evident. Each legend line in the plot corresponds to a muscle activation channel in the gait cycle, which is used for automating healthy control subjects’ movements. Notably, healthy control subjects exhibited peak levels of up to 240 millivolts, which were followed by a sharp decline and reflecting changing muscle activation patterns over time. Conversely, in Figure 6, the gait pattern of patients reveals lower amplitude muscle activation compared to healthy controls. Each legend in Figure 6 also represents the muscle activation channel derived from the gait cycle in the EMG data, which is employed in automating patient movement patterns. A clear distinction emerges between the higher amplitudes observed in healthy control subjects and the lower amplitudes in patients. This comparison underscores the altered muscle activation pattern in the gait cycle of patients, which is indicative of distinct movement patterns from those of healthy control subjects.

## 4. Discussion

Gait analysis provides valuable information on how patients with polymyalgia rheumatica disease walk. This study presents a clinical gait classification of patients with polymyalgia rheumatica versus control subjects. Four classification models were used to classify the gait patterns for the two groups. The main highlights of the results are the (i) classification of the clinical gait pattern in patients with polymyalgia rheumatica based on the strained hip muscles and (ii) discrimination of the movement for normal and impaired gait patterns of the two groups. This study found that SVM recorded the highest accuracy of 85%, while RF had an accuracy of 80%. KNN had an accuracy of 75% while DT recorded the lowest accuracy of 70%. In terms of sensitivity, SVM recorded 92%, RF had 86%, KNN recorded 84% and DT had a sensitivity of 90%. SVM and RF had the highest sensitivity of 86% in discriminating between normal and impaired gait patterns. From the results, SVM had the highest specificity of 71% while DT had the lowest of 50%.

Some studies have investigated pathological and physiological features in classifying human movement with kinematic data [27,28]. Although there are some advantages of using kinematic data for gait analysis, kinematic data are more difficult to obtain. This accounts for the reasons why EMG data are desirable for the classification of gait disorders. In addition, clinical scales are used to assess gait rather than movement analysis systems because 3D motion analysis systems have limitations in terms of the variation and poor granularity of kinematic data. EMG data were suitable for this study to examine the superficial hip muscle activity of patients and healthy controls during the gait cycle. The EMG data provided additional features and information on the neurobiological underpinning of gait features, which was useful in abnormal gait classification.

Comparing this study with recent works in [10,18] that used EMG data for detecting gait abnormality, there are significant differences. Firstly, this study focused on polymyalgia rheumatica, which is a common autoimmune muscular disorder. This is essential in investigating the effects of the strained hip muscles on the movement of patients. This provides an understanding of the functional implications of the hip muscles in the early detection of impairment. This is significantly different from previous studies on movement analysis of patients with musculoskeletal disease. This is the first study to carry out this investigation on the functional effect of strained hip muscles on PMR patients to the best of our knowledge. Secondly, in this study, we used four specific hip muscles affected by the disease. These were the dominant hip muscles affected by the disease, which play an essential role in movement coordination. In addition, contrasting this study with the research conducted by Fei et al. [13] underscores certain distinctions. While this study examines the muscle synergies of PMR patients and healthy control subjects, Fei’s work explores differences in muscle synergies between CP children and adults during walking, emphasizing variations in both the number and structure of synergies. Also, this study aimed to analyze the gait of PMR patients by focusing on specific hip muscle synergies to detect abnormalities in walking patterns. In contrast, Barroso et al. [15] sought to enhance the evaluation of walking in post-stroke hemiparetic patients. They combined muscle synergies and biomechanical analysis, assessing both lower limbs to predict impaired gait using the Fugl–Meyer Assessment score.

There is evidence by Chakraborty et al. [12] to suggest that a smaller number of muscle data are sufficient in distinguishing between healthy and impaired gait patterns. Even though the analytical approach used is similar, this study used an EMG data extraction process that originated from the motor neuron measurement of patients. Patients’ hip muscle pain may lead to an inhibition of muscle activation and changes in patterns to protect the affected area. Additionally, hip muscle inflammation also interferes with the normal functioning of the muscular system and impairs muscle activation compared to healthy control subjects. Depending on the musculoskeletal dysfunction, individual patients may employ alterations in compensatory movement patterns and muscle activation strategies to reduce pain. It is important to note that these factors may not be exclusive and could vary depending on underlying conditions.

In discriminating between patients and healthy control subjects based on their movement, this work utilized muscle activation patterns to distinguish different phases of the gait cycle. We found that patients showed delays in the beginning and early endings of muscle activation compared to healthy individuals. The disparity in the timing of muscle activation onset and offset between the two groups suggests altered hip muscle activation in patients. Irregular patterns of muscle activation, such as prolonged activation, indicate potential impairments in the gait cycle. Additionally, patients displayed lower EMG signal amplitudes compared to healthy controls, which could indicate muscle weakness. Automated gait patterns for both groups are depicted in Figure 6 and Figure 7. A noticeable variation in the gait cycle of patients compared to controls was observed with controls showing slightly higher amplitudes. This suggests a difference in the role of biomechanical factors in walking between PMR patients and healthy controls. The strained hip muscles in patients may contribute to movement impairments, although this conclusion is not definitive, as other factors could also influence gait impairments in PMR patients.

One limitation of this work is the number of participants recruited for the study. Even though we aimed for 35 patients, only 18 were recruited for the period. Another limitation was the unbalanced classes between patients and healthy control subjects. Nevertheless, we were able to obtain sufficient data for analysis. Our results demonstrate that machine learning models could increase the usefulness of clinically available data to classify movement disorders. The potential applications of the trained models could offer valuable information in classifying and analyzing various movements, making them essential tools for tasks such as gait analysis and sports science. Future studies would apply deep learning techniques for the clinical gait classification of PMR disease.

## 5. Conclusions

Gait analysis can be used as an essential tool to monitor PMR disease progression over time. In this study, we presented an automatic gait classification to discriminate between the gait patterns of PMR patients and healthy control subjects. We found out that SVM and RF classifiers performed satisfactorily with accuracies of 85% and 80%, respectively. KNN recorded an accuracy of 75%, while DT was the least accurate, performing with 70% accuracy. The approach in this study appears to be satisfactory and could be used in a large dataset for the classification of gait disorders for patients with similar conditions. The results of this study may be used by clinicians in designing therapeutic exercises as an integral part of a treatment plan. Our findings may also be used to design a decision support system for diagnostic purposes.

## Figures and Tables

**Figure 1 sensors-24-01500-f001:**
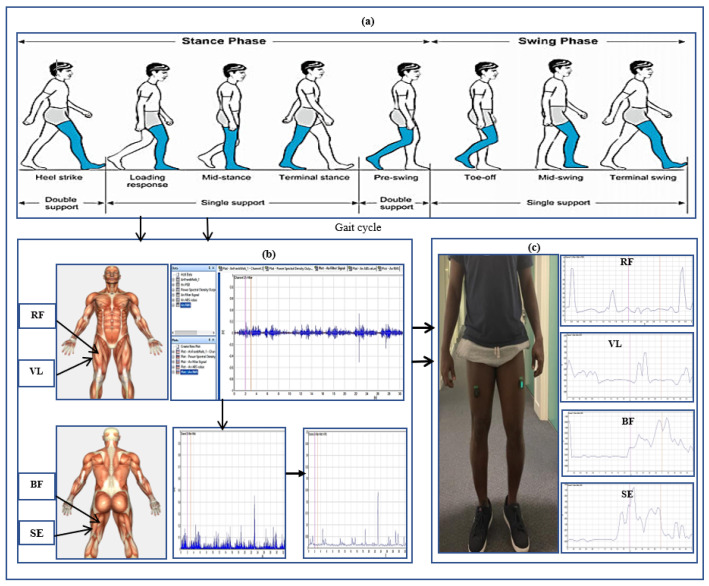
A cluster diagram illustrating (**a**) the gait cycle (**b**) the recorded signal processing of specific hip muscles and (**c**) a participant engaged in the gait exercise with enveloped EMG signal displayed.

**Figure 2 sensors-24-01500-f002:**
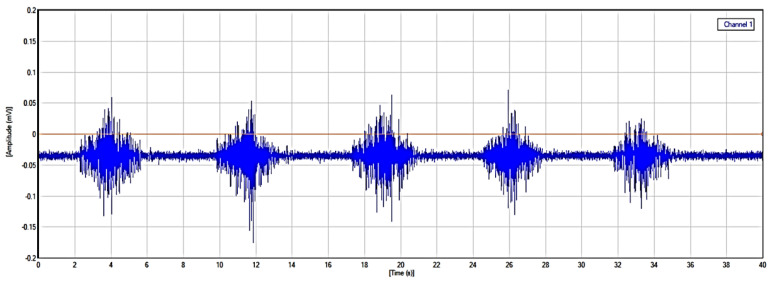
Raw EMG signal.

**Figure 3 sensors-24-01500-f003:**
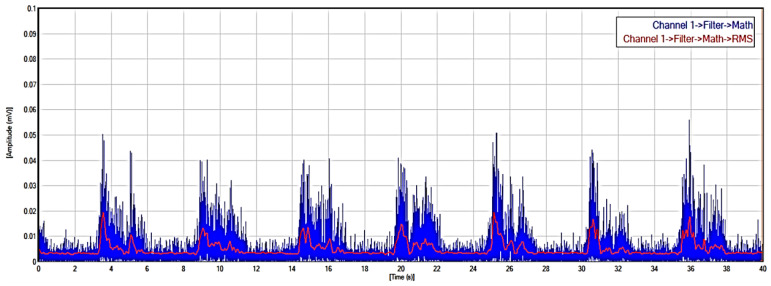
Filtered and rectified EMG signal.

**Figure 4 sensors-24-01500-f004:**
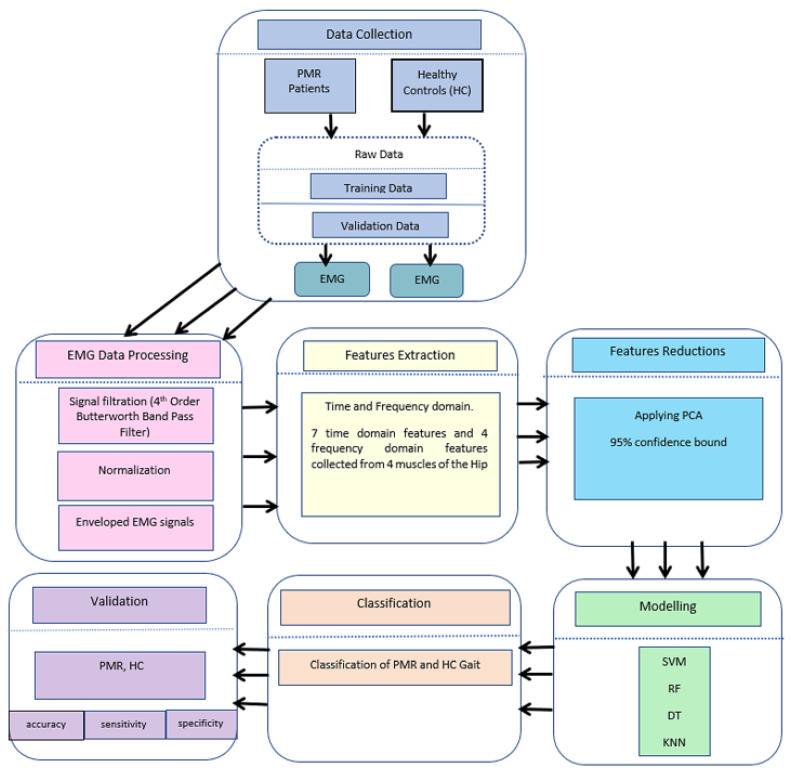
Flow chart for the classification process.

**Figure 5 sensors-24-01500-f005:**
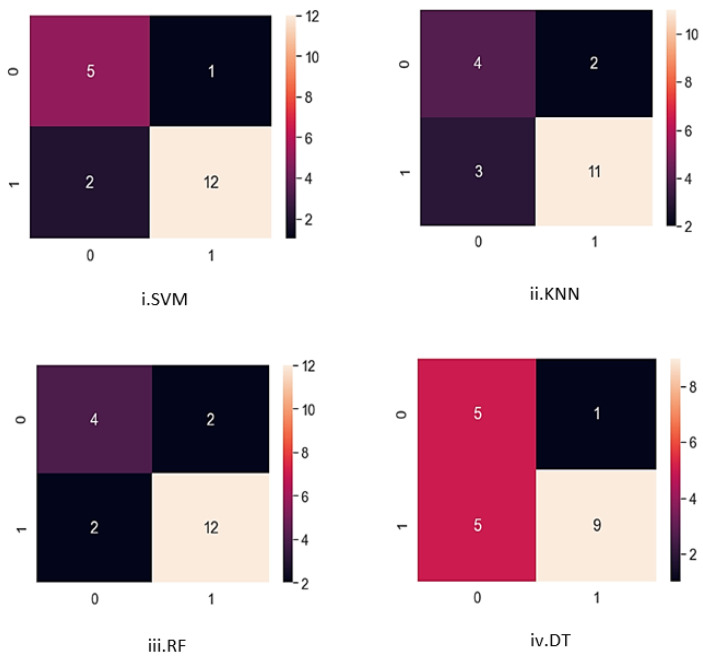
Confusion matrix plots for the classification algorithms.

**Figure 6 sensors-24-01500-f006:**
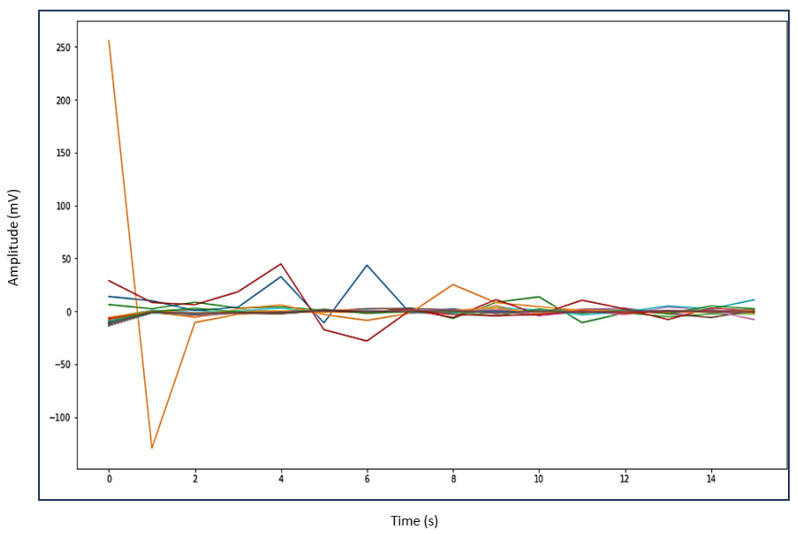
Patients gait pattern.

**Figure 7 sensors-24-01500-f007:**
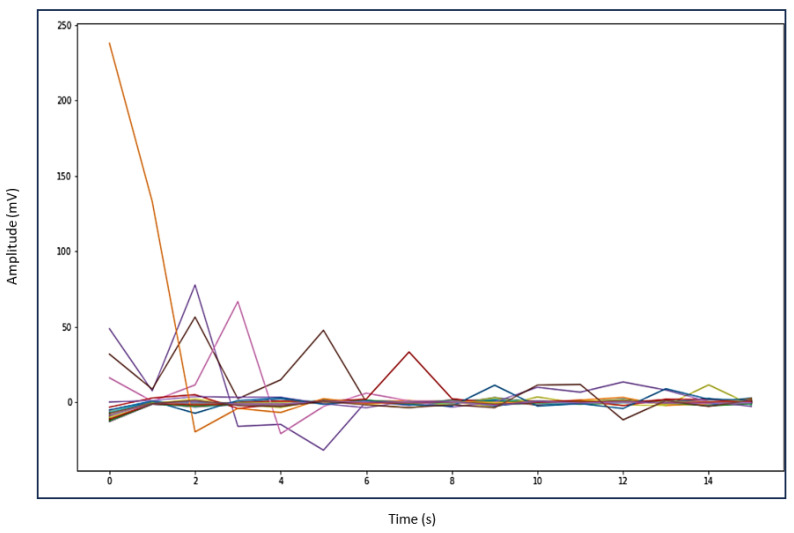
Healthy control subjects gait pattern.

**Table 1 sensors-24-01500-t001:** Features extracted from EMG data.

Extrated Features in Time and Frequency Domain	Formulae Definition
Root Mean Square (RMS)	$\sqrt{\frac{1}{N}*\sum_{i=1}^{N}EMG^{2}}$
Mean Absolute Value (MAV)	$\frac{1}{N}\sum_{i=1}^{N}\left|nEMG\right|$
Integrated EMG (IEMG)	$\sum_{i=1}^{N}\left|EMG_{i}\right|$
Simple Square Integral (SSI)	$\sum_{i=1}^{N}nEMG_{i}^{2}$
Variance of EMG (VAR)	$\frac{1}{N-1}\sum_{i=1}^{N}nEMG_{i}^{2}$
Modified Mean Abs Value (MMAV)	$\frac{1}{N}\sum_{i=1}^{N}\omega_{i}\left|nEMG_{i}\right|$
Avg.Amplitude Change (AAC)	$\frac{1}{N}\sum_{i=1}^{N}\left|nEMG_{i}+1-nEMG_{i}\right|$
Mean Frequency (MNF)	$\frac{\sum_{j=1}^{M}f_{j}W_{j}}{\sum_{j=1}^{M}W_{j}}$
Median Frequency (MDF)	$\sum_{j=1}^{medFreq}W_{j}=\sum_{j=medFreq}^{M}W_{j}$
Mean Power (MNP)	$MNP=\frac{\sum_{j=1}^{M}W_{j}}{M}$
Total Power (TP)	$\sum_{j=1}^{M}W_{j}=sm0$

From the formulae definition in Table 1, EMG denotes the amplitude of the muscle signal at a specific time point *i* and *n* represents the number of EMG amplitudes, while *N* indicates the total number of sample points in the EMG signal. For the frequency domain, $f_{j}$ refers to the frequency spectrum at a particular frequency bin *j* and $W_{j}$ stands for the power at that frequency. *M* represents the number of sample points at that frequency and *j* signifies the frequency points of the signal. Lastly, sm is the power spectrum moment of signal in the frequency domain.

**Table 2 sensors-24-01500-t002:** Classification performance of the machine learning algorithms in a 10-fold cross-validation.

Classifier	Accuracy	Sensitivity	Specificity
SVM	85%	92%	71%
KNN	75%	84%	57%
DT	70%	90%	50%
RF	80%	86%	67%

**Table 3 sensors-24-01500-t003:** EMG signal plot for gait cycle.

EMG Signal from Muscle	Patients	Healthy Control Subjects
(a) Onset and offset	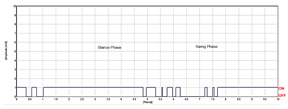	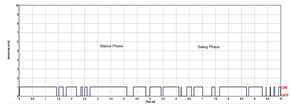
(b) Rectus femoris	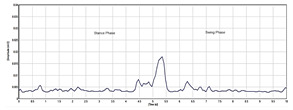	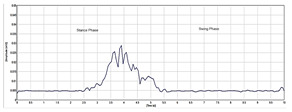
(c) Vastus lateralis	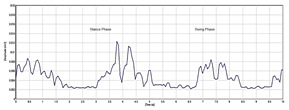	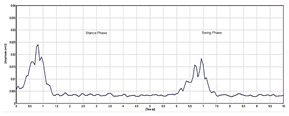
(d) Biceps femoris	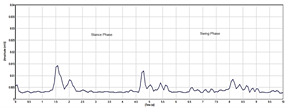	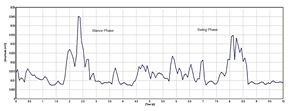
(e) Semitendinosus	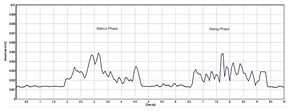	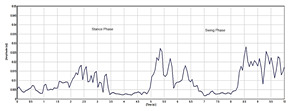

## Data Availability

Data can be made available upon request to the relevant institution.

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
