# Peer review of "A Movement Classification of Polymyalgia Rheumatica Patients Using Myoelectric Sensors"

_sensors, 2024, doi:10.3390/s24051500_

Round 1
Reviewer 1 Report
Comments and Suggestions for Authors
Please provide the number of healthy and sick people in both groups: 18 people and 7 people. The group is too small for the results to be considered fully convincing. Furthermore, it is likely that the data is not balanced. Please comment on this issue.
On which group of data were the classifiers trained? The heading of table 2 mentions ten-fold cross-validation, so the number of people taken into account should be divisible by 10. Please explain, because the phrase control group is misleading.
Figures 4 and 5 are not understandable (what the colors mean?), nor is their explanation.
The best result of the SVM classifier is not surprising.
Did the Authors try to fuse the four classifiers used?
Line 46: BiLSMT –-> BiLSTM
Tab. 1: Please explain nEMG, why sum with respect to n?
Line 187, 194: what does “automate the movement” mean?
Author Response
Responses to reviewer one comments is attached in a pdf file.

Reviewer 2 Report
Comments and Suggestions for Authors
The article "A Movement Classification of Polymyalgia Rheumatica Patients
Using Myoelectric Sensors" - a good attempt to obtain data for an expert system using local material.
There are some points that could improve this work, as well as avoid the authors from making unjustified generalizations.
Table 1. You have all the classic EMG analysis parameters and even more. However, there are completely no parameters characterizing the phase (relative to the gait cycle) of maximum muscle activity. Let me clarify the idea - all the muscles analyzed in this study have one maximum activity at the beginning of the gait cycle. This is only for healthy people. But in patients it tends to shift in time. This is most pronounced for another category of diseases. But besides the change (decrease) in amplitude, there is another dimension - in time. It is the moment when maximum activity occurs. This parameter, a very important one, was not analyzed in this work. But if the time of onset of maximum muscle activity changes, then this changes much more than a decrease (change) in its maximum, average, integral, median... amplitude. I'm not pushing for this to be changed right now. Too significant changes, however it would be right. It's unlikely that you are ready for this. But I would like to know why you did not take this into account in this work?
Line 213-215 – This statement about the more frequent use of EMG is true, but only for the reason that all other parameters, and especially kinematics, are technically more difficult to obtain. This is also the reason why clinical scales are used to assess gait rather than movement analysis systems.
Line 216-220 – Regarding this statement, this is partly true. EMG technology allows (non-invasively) to examine only superficial muscles. And the kinematics are affected not only by the surface ones, but also by all the others. For example musculus iliopsoas plays a very significant role in hip flexion, but cannot be studied by EMG.
Author Response
Responses to reviewer two comments is attached in a pdf file.

Reviewer 3 Report
Comments and Suggestions for Authors
The manuscript proposes using EMG data to classify patients with gait disorders from healthy ones. Several classifiers are investigated for this purpose including SVM, RF, kNN, and DT. Experimental results are analyzed by comparing the accuracy and sensitivity achieved by different classifiers.
--The task is formulated as a binary classification problem discriminating patients from healthy subjects. What are the motivations for such a formulation? What are the potential applications of these trained models? The authors should clarify why this is a meaningful problem to solve in the first place.
--The description of classifiers needs to be improved by adding more technical details as well as the hyper-parameters involved. In particular, the less typical classifier Rotation Forest deserves more introduction, and why it was chosen in this study.
--How are the data split into training and test subsets? It is worth considering the strategy of splitting data subject-wise, i.e., data from the same subject should fall into either the training or the test subset. This is to investigate the generalization of the trained classifier.
--It is suggested to investigate the importance of different features in this task.
--The whole data processing pipeline should be presented with details, e.g., denoising, segment detection, etc.
--The authors are suggested to refer to related works [1-2] to improve the study and manuscript.
[1] Lower-limb muscle synergies in children with cerebral palsy
[2] Acceleration and electromyography (EMG) pattern recognition for children with cerebral palsy
Comments on the Quality of English LanguageThe English language is fine.
Author Response
The manuscript proposes using EMG data to classify patients with gait disorders from healthy ones. Several classifiers are investigated for this purpose including SVM, RF, kNN, and DT. Experimental results are analyzed by comparing the accuracy and sensitivity achieved by different classifiers.
--The task is formulated as a binary classification problem discriminating patients from healthy subjects. What are the motivations for such a formulation? What are the potential applications of these trained models? The authors should clarify why this is a meaningful problem to solve in the first place.
Our motivation for this binary classification in electromyography (EMG) datasets for clinical gait analysis lies in the ability to classify gait patterns based on hip muscle activity. Surface electromyography (sEMG) signals were suitable for analyzing PMR disease due to their ability to provide valuable insights into the muscle function of patients. Furthermore, EMG allows for the classification of different gait periods, such as Stance and Swing, which is essential for understanding muscle activation patterns. This classification was valuable in monitoring PMR patients as it provides insights into the dynamic activity of muscles during pathological gait and helps identify abnormal walking patterns.
The potential applications of the machine learning training with EMG data, offer valuable applications in classifying and analyzing various movements, making them essential tools for tasks such as gait analysis, sports science, and musculoskeletal disorder diagnosis. This could be essential in the real-time classification of patients with different conditions.
--The description of classifiers needs to be improved by adding more technical details as well as the hyper-parameters involved. In particular, the less typical classifier Rotation Forest deserves more introduction, and why it was chosen in this study.
The classification of electromyography (EMG) data for patients with polymyalgia rheumatica (PMR) is a meaningful problem to solve and provide valuable insights into the hip muscle activity and function of patients with PMR. This can provide useful information to support clinicians in the monitoring of patients. By classifying EMG data, it becomes possible to identify specific patterns associated with PMR. Furthermore, gait classification can be used to monitor the response to treatment and the progression of PMR. This can help healthcare professionals assess the effectiveness of interventions. This has been explained in the manuscript.
--How are the data split into training and test subsets? It is worth considering the strategy of splitting data subject-wise, i.e., data from the same subject should fall into either the training or the test subset. This is to investigate the generalization of the trained classifier.
The description of the classifiers has been improved in the manuscript providing technical details and key parameters for each classifier.
--It is suggested to investigate the importance of different features in this task.
The study uses a subject-wise to split the data for training and testing for each of the classifiers used. This has been included in the manuscript.
--The whole data processing pipeline should be presented with details, e.g., denoising, segment detection, etc.
The data processing pipeline has been presented with more details illustrating the EMG signal processing and segment detection.
--The authors are suggested to refer to related works [1-2] to improve the study and manuscript.
[1] Lower-limb muscle synergies in children with cerebral palsy
[2] Acceleration and electromyography (EMG) pattern recognition for children with cerebral palsy
The related works have been considered to help improve this manuscript.
Round 2
Reviewer 1 Report
Comments and Suggestions for Authors
The new version of the paper is better but please consider two additional comments:
1. Ten-fold cross-validation. Please do not be surprised when a reader is interested in a more challenging protocol, i.e., the loso (leave-one-subject-out). Here a tested person does not appear in the training set. Moreover, there are readers who want to repeat your experiment. So they would like to get precise information about the division of your data into ten parts.
2. Table 1. When I see a summation with respect to an index i, I expect that the argument depends on i. This is not shown in your presentation (see, e.g., lines 1, 2). Moreover, when I see a sum with respect to n and the argument nEMGi I consider it as confusing. Please add an explanation what are N, n, i and correct the table if necessary.
Author Response
Responses to reviewer one uploaded in a file.

Reviewer 3 Report
Comments and Suggestions for Authors
The revised manuscript has addresses most of my concerns, however, there still lack some essential references in the introduction and discussion sections as a journal paper.
Author Response
Responses to reviewer three uploaded in a file.
